# Cerebral Seizures in an Adolescent with Jervell and Lange-Nielsen Syndrome: It May Not Be Epilepsy

**Joachim Levaux** [1], **Nesrine Farhat** [1], **Lieve Van Casteren** [2], **Saskia Bulk** [3] **and Marie-Christine Seghaye** [1,*]

1    Department of Pediatric Cardiology, University Hospital Liège, 4000 Liège, Belgium
2    Department of Cardiology, University Hospital Liège, 4000 Liège, Belgium
3    Department of Genetics, University Hospital Liège, 4000 Liège, Belgium
*    Correspondence: mcseghaye@chuliege.be; Tel.: +32-4-323-9275

**Abstract:** A 13-year-old girl with Jervell and Lange-Nielsen syndrome associated congenital long QT syndrome (LQTS) and central deafness was admitted for generalized seizures. LQTS had been diagnosed after birth and confirmed at genetic testing. β-blocker treatment was immediately started. Despite this, since the age of 12 months, recurrent cerebral seizures occurred leading to the diagnosis of epilepsy. Anti-convulsive therapy was initiated but without success. At the last admission, nadolol dosage seemed infratherapeutic. Considering malignant ventricular arrhythmias as the cause of seizures, the β-blocker dosage was adjusted to weight and levels of magnesium and potassium optimized. Furthermore, the patient received an implantable Medtronic Reveal LINQ Recorder®. Since then, the adolescent has been asymptomatic with no arrhythmia documented. LQTS is due to one or more mutations of genes coding for ion channels. It may induce malignant ventricular arrhythmias and is a major cause of sudden cardiac death in children. Generalized cerebral seizures are extra-cardiac manifestations caused by decreased cerebral perfusion during ventricular arrhythmia. They are commonly misinterpreted as manifestations of epilepsy. For any patient with known or unknown LQTS who presents seizures with resistance to anti-convulsive therapy, a cardiac electrophysiological investigation should be performed promptly to ensure etiological diagnosis and optimize treatment.

**Keywords:** congenital long QT syndrome; Jervell and Lange-Nielsen syndrome; genetics; ventricular arrhythmia; seizures; beta-blocker

## 1. Introduction

Congenital long QT syndrome (LQTS) has an incidence of 1/2500 living birth and is due to a mutation of one or several genes coding for ion channels inducing prolongation of the QTc interval on standard ECG. It carries a high risk for malignant ventricular arrhythmias and sudden cardiac death [1–4]. LQTS may be associated with central deafness in the rare case of Jervell and Lange-Nielsen syndrome (JLN) [1,4,5].

Malignant ventricular arrhythmias may manifest as syncope with or without seizures, a clinical picture that usually leads to the diagnosis of epilepsy. Commonly, the link between channelopathy and neurological manifestations is unknown to pediatricians, rendering the correct diagnosis delayed given the patient survives.

The clinical case reported here is aimed at increasing awareness of physicians caring for children presenting atypical forms of epilepsy and who are at risk to die suddenly in the absence of adequate anti-arrhythmic prophylaxis.

## 2. Clinical Case

A 13-year-old female adolescent with JLN diagnosed early after birth was admitted to the emergency room for general seizures. Genetic exploration had shown the presence of a heterozygotous SCN5A mutation (c.1715 c > 4) and a yet not described homozygotous KCNQ1 mutation (IVS4+5G>A). This latter mutation was also found in a heterozygotous

state in her mother, her sister and her half-brother who all show LQTS without deafness Figure 1 depicts the family tree.

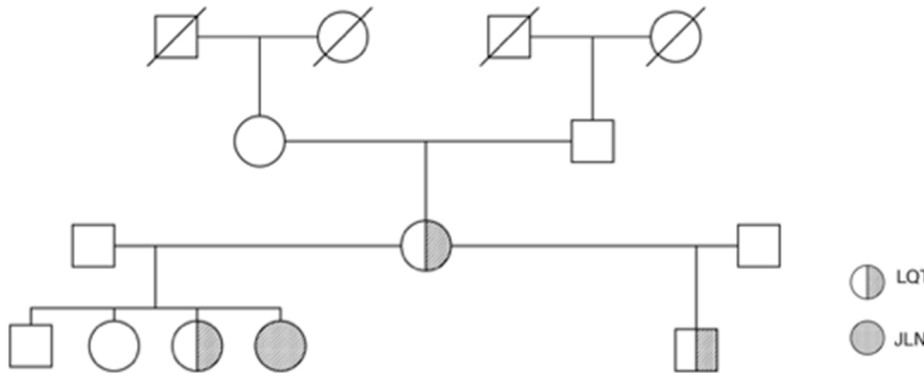

**Figure 1.** Patient family tree.

Our female patient with Jervell and Lange-Nielsen syndrome, homozygotous KCNQ1 mutation and heterozygotous SCN5A mutation is depicted as a full circle (JLN). Her mother, sister and half-brother with heterozygotous KCNQ1 mutation are depicted as half-shadowed circles and squares, respectively.

In our patient, the duration of QT corrected for the heart rate by the Bazett formula was repeatedly reported to exceed 500 ms. T wave alternans, a periodic beat-to-beat variation in the amplitude or shape of the T wave were present (Figure 2).

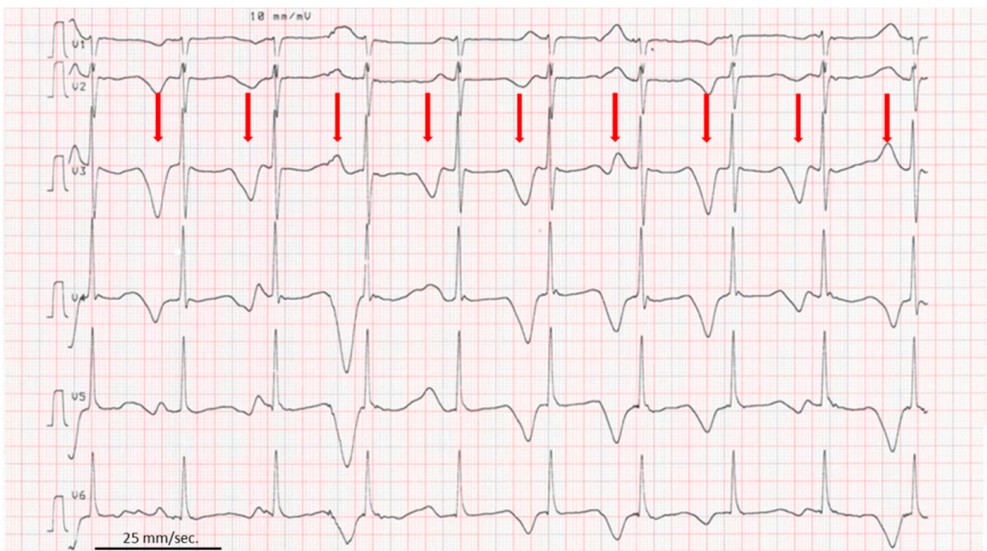

**Figure 2.** ECG showing long QT (QTc: 610 ms.) and T wave alternans (red arrows). Recording speed is 25 mm/s.

β-blocker prophylaxis (propranolol 1 mg/kg/day) was started in the neonatal period and given without any interruption.

The parents were advised early that the child would have to strictly avoid lifelong competitive sports in the unlikely case exercise tolerance under β-blocker treatment would be normal. Children's play activities were allowed.

At the age of 12 months, the patient presented the first generalized cerebral seizures in the context of severe hyponatremia occurring after cochlear implantation. Treatment by phenobarbital (5 mg/kg/day) was started but several episodes of generalized seizures occurred in the following months. Diagnosis of epilepsy was made despite negative neurological exploration including two standard and one prolonged night electro-encephalography, and one cerebral computed tomography.

Treatment with phenobarbital was finally stopped by the parents at the age of 3 years. Between the age of 3 and 9 years, the patient presented four more episodes of epileptic equivalents without abnormal movements but with wetting and loss of consciousness. The EEG performed after these syncopes were all normal, thus anti-epileptic treatment was considered not indicated. At the age of 9 years, the patient underwent a new routine cardiological exploration with standard- and Holter-ECG, chest X-ray (Figure 3) and echocardiography to exclude any structural cardiac anomaly. Propranolol was then replaced by nadolol, which was continued until the last admission to our institution at the age of 13 years. The patient was lost to cardiological follow-up for the last years with the latest prescribed nadolol dosage of 1.25 mg/kg/day. Adequate compliance with the medical treatment was uncertain.

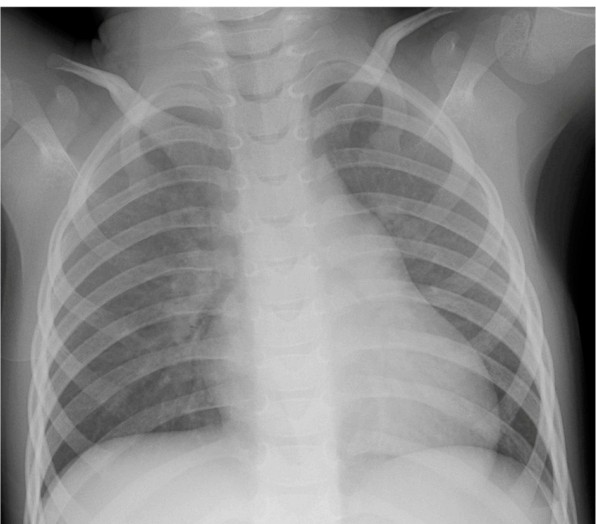

**Figure 3.** Chest X-ray showing normal cardiac size.

The cardiologic work-up led to suspect malignant tachyarrhythmia. Consecutively, nadolol dosage was increased (1.7 mg/kg/day), levels of potassium and magnesium optimized by a treatment of spironolactone (1 mg/kg/day) and magnesium (7.5 mg/kg/day), respectively. Finally, an implantable reveal device (Medtronic Reveal LINQ Recorder®) was placed for tele-monitoring. Under the optimized medical treatment, the patient did not show any new episodes of seizures. After a follow-up period of two years, there were no ventricular arrhythmias recorded.

## 3. Discussion

LQTS is due to the dysfunction of ion channels involved in myocardial cell repolarization. Several mutations of genes coding mainly for ion channel sub-units have been identified (Table 1) that lead to increase action potential duration and QT interval prolongation [1,2,4,6,7].

LQTS usually follows an autosomal dominant inheritance pattern as is the case of LQT1 to LQT15. The most frequent forms (90%) of LQTS are LQT1, LQT2 and LQT3, which involve mutations of the genes coding for KCNQ1, KCNH2, and SCN5A, respectively [1,2,8].

JLN associates long QT with bilateral central perception deafness and represents one of the most severe forms of LQTS [1–3,9]. It is an autosomal recessive inherited disease characterized by bi-allelic mutations of the genes KCNQ1 (JLN1) or KCNE1 (JLN2) [1] responsible for internal ear current reduction in addition to prolongation of the myocardial cell action potential [2].

**Table 1.** The most frequently involved genes in Long QT syndrome (LQTS). The mutation frequency shown is that for the LQTS population (%).

| Type of LQTS | Gene | Mutation Frequency | Locus |
|---|---|---|---|
| LQT1 | KCNQ1 | 40–55 | 11p15.5 |
| LQT2 | KCNH2 | 30–45 | 7q35-36 |
| LQT3 | SCN5A | 5–10 | 3p21-24 |
| LQT4 | ANKB | <1 | 4q25-27 |
| LQT5 | KCNE1 | <1 | 21q22.1 |
| LQT6 | KCNE2 | <1 | 21q22.1 |
| LQT7 | KCNJ2 | <1 | 17q23 |
| LQT8 | CACNA1C | <1 | 12p13.3 |
| LQT9 | CAV3 | <1 | 3p25 |
| LQT10 | SCN4B | <1 | 11q23.3 |
| LQT11 | AKAP9 | <1 | 7q21-22 |
| LQT12 | SNTA1 | <1 | 20q21-22 |
| LQT13 | KCNJ5 | <1 | 11q24 |
| LQT14 | CALM1 | <1 | 14q32.11 |
| LQT15 | CALM2 | <1 | 2p21 |
| **Jervell et Lange-Nielsen Syndrome** | | | |
| JLN1 | KCNQ1 | <1 | 11p15.5 |
| JLN2 | KCNE1 | <1 | 21q22.1-22.2 |

### 3.1. Clinical Presentation

LQTS manifests clinically by malignant ventricular tachyarrhythmia, particularly by *torsades de pointes* that often resolve spontaneously in contrast to classical ventricular fibrillation. LQTS therefore may manifest by syncope due to decreased cerebral blood flow with or without cerebral seizures. Generalized seizures misinterpreted as manifestations of epilepsy may be the only modus of presentation of LQTS as was the case in our patient. A previous report of a 5-year-old child with refractory epilepsy in whom the diagnosis of JLN was finally made and in whom β-blocker treatment led to the resolution of the neurological manifestations supports our view about the need to render the medical community aware of the link existing between LQTS and recurrent cerebral seizures misleading to the diagnosis of refractory epilepsy [10].

Importantly, *torsades de pointes* may end up in ventricular fibrillation and sudden cardiac death may be the only event in patients with LQTS [1,4,6,7].

In our patient, the diagnosis of epilepsy was retrospectively revised because none of the numerous EEGs performed over time had shown any anomaly. Furthermore, the patient was free from neurological manifestations as soon as the medical prophylaxis of the ventricular arrhythmias was optimized.

### 3.2. Electrocardiography

LQTS is characterized by a pathologic prolongation of QT duration measured on the standard electrocardiogram in DII or V5 as the interval between the beginning of the QRS complex and the end of the T wave, a juvenile U wave being excluded.

QT interval duration is influenced by numerous factors including those related to the recording- and measurement technique [6]. It must be corrected for heart rate [8,11]. Usually, the corrected QT (QTc) interval for heart rate is calculated by the Bazett formula where $QTc = QT/\sqrt{R-R}$.

QT interval duration >450 ms is related to a high probability of LQTS in men and >460 ms in women [2,4,5]. In children, no gender differences are present. There is a

physiological prolongation of QTc by the second-month of life followed by a progressive decrease so that by 6 months of age QTc returns to the values recorded in the first week of $400 \pm 20$ ms [12,13]. Besides QT interval duration, T wave morphology may be helpful to detect long QT: bi-phasic T wave or T wave *alternans* are suggestive of LQTS (Figure 2) [2].

### 3.3. Diagnosis of LQT

The patient's probability to show LQTS requires to be evaluated on a clinical and electrophysiological basis.

The Schwartz score allows estimating patient risk to present LQTS. This score is based on clinical and electrophysiological findings in the patient and her/his family (Table 2) [1,11]. The Schwartz score has a high specificity (99%) but a low sensibility (18%) [8].

**Table 2.** Schwartz score diagnostic criteria for Long QT Syndrome (LQTS).

| Electrocardiographic Findings | Score |
|---|---|
| A. QTc (Bazett) | 3 |
| • $\geq$480 ms | |
| • 460–479 ms | 2 |
| • 450–459 ms (in males) | 1 |
| B. QTc fourth minute of recovery form exercise test $\geq$ 480 ms | 1 |
| C. Torsade de Pointes | 2 |
| D. T waves alternans | 1 |
| E. Notched T wave in 3 leads | 1 |
| F. Low heart rate for age | 0.5 |
| **Clinical History** | **Score** |
| A. Syncope | |
| • with stress | 2 |
| • without stress | 1 |
| B. Congenital deafness | 0.5 |
| **Family History** | **Score** |
| A. Family members with definite LQTS | 1 |
| B. Unexplained sudden cardiac death below age 30 among immediate family members | 0.5 |

Ranging from 0 to 9, a score value $\leq$1 indicates low probability-, a score between 1 and 3 indicates intermediate probability- and a score $\geq$3.5 indicates a high probability for the patient to carry LQTS. The Schwartz score allows the selection of patients who are eligible for further investigations such as Holter-ECG, exercise ECG, Tilt-test, or epinephrine stress test. The latter test allows unmasking LQTS in patients who present normal QT interval at rest with a predictive value of 75% [14].

Genetic testing is recommended in any patient with a Schwarz score $\geq$3.5 [11] and in her/his first-degree relatives. The presence of a pathogenic mutation of a gene coding for an ion channel allows diagnosing LQTS in an even asymptomatic individual [1,2,4]. Conversely, the absence of such a mutation does not exclude LQTS in a patient with a positive phenotype [11].

The documentation of an episode of ventricular arrhythmia or *torsade de pointes* might be challenging if spontaneous resolution occurs in the absence of monitoring. In case of suspicion of malignant ventricular arrhythmia, implantation of a sub-cutaneous monitoring device must be considered in order to correlate clinical events such as syncope or seizures with electrocardiographic anomalies.

In our patient, implantation of the reveal device was simultaneous to optimization of medical treatment, in particular β-blocker administration. The fact that seizures disappeared after this and that no more cardiac event was recorded suggest that seizures were secondary to malignant tachy-arrhythmias [15].

According to HRS/EHRA/APHRS expert consensus, LQTS is diagnosed:

- In the presence of an LQTS risk score ≥3.5 in the absence of a secondary cause for QT prolongation and/or;
- In the presence of an unequivocally pathogenic mutation in one of the LQTS genes or;
- In the presence of a QT interval corrected for heart rate using Bazett's formula (QTc) ≥500 ms in repeated 12-lead ECG and in the absence of a secondary cause for QT prolongation [16].

LQTS can be diagnosed in the presence of a QTc between 480–499 ms in repeated 12-lead ECGs in a patient with unexplained syncope in the absence of a secondary cause for QT prolongation and in the absence of a pathogenic mutation [16].

### 3.4. Treatment of LQTS

The treatment of LQTS is aimed at preventing the development of ventricular tachyarrhythmia and sudden cardiac death due to the sudden activation of the sympathetic activity mediated by the left cardiac stellar ganglion. Lifestyle modifications such as avoidance of drugs that prolong the QT interval and strenuous exercise should be systematic in all patients with LQTS. Participation of LQTS patients in competitive sports is still debated among experts [16].

All patients (particularly those with LQT2) need to maintain sufficient cellular potassium concentrations by optimal food intakes and, if necessary, by potassium-sparing diuretics such as spironolactone or potassium supplementation.

- *β-Blockers*

β-blockers are the treatment of the first choice in patients with LQTS and unless the patient has contra-indications. They are generally well tolerated and clinically indicated in LQTS, including those with a genetic diagnosis and normal QTc [1,2,4,7,16].

β-blockers do not all have the same efficacy in terms of prophylaxis of malignant ventricular tachyarrhythmia in patients with LQTS [17]. Propranolol and nadolol, both non-cardio-selective molecules are the most efficient. In contrast to propranolol, which must be administrated in 3–4 doses per day, nadolol, with its longer half-life, only requires 1-2 doses per day. This improves patient compliance, especially in children [2,7,11].

β-blockers have been shown to be very effective in patients with LQT1, given the compliance to the treatment is good and the patient does not take any substance increasing QT interval [2].

β-blockers are less effective in patients with LQT2 et LQT3 [2,11]. In case of persistence of life-threatening events and despite dose-adapted non-cardio-selective β-blocker treatment, other therapeutic means must be considered, particularly in patients with JLN [2]. In our patient with JLN, optimization of cardio-selective β-blocker treatment and adjuvant treatment by spironolactone and magnesium supplementation led to effective adverse event prophylaxis for at least 2 years.

- *Implantable Cardiac Defibrillator*

There is general consensus about the indication of an implantable cardiac defibrillator (ICD) in any patient having presented sudden cardiac death regardless of the medical treatment status. ICD can be implanted even in young infants. ICD should not be used as first-line therapy in asymptomatic patients and rather be favored in patients with LQTS-related syncope despite optimal β-blocker treatment. Nevertheless, ICD may be considered in patients who are deemed to be at very high risk (especially those with a contraindication to beta-blocker therapy) [16].

A new score, the 1-2-3-LQTS-Risk, has been proposed to aid clinicians to identify patients at the highest risk of life-threatening arrhythmias who could particularly benefit from an ICD [18].

- *Left Cardiac Sympathetic Denervation*

Left cardiac sympathetic denervation (LCSD) consists of the ablation of the first 3-4 thoracic sympathetic nodes (the cephalic portion of the left stellate ganglion is left intact to avoid the Horner syndrome). The indication must be discussed in any patient presenting recurrent syncope due to malignant ventricular tachyarrhythmia despite optimally conducted medical treatment [1,2,4,7,11]. In refractory cases, a tri-therapy associating β-blocker treatment, LCSD and ICD might be proposed [2,4,7,11].

### 3.5. Clinical Implication of the Genotype-Phenotype Correlation

Recent important progress has been made allowing a better understanding of the genotype-phenotype relationship in patients with LQTS [2].

Patients with LQT1 encounter a higher risk to develop malignant ventricular arrhythmia in case of sympathetic activation such as exercise or strong emotions. They therefore should not participate in high-level sports competitions and should avoid swimming in the absence of supervision due to the high risk of drowning [2,4,19]. Our patient showing a homozygote mutation of the KCNQ1 gene followed recommendations for LQT1 patients.

LQT2 patients are highly sensitive to acoustic stimuli and emotional stress. Therefore, telephone devices and alarm clocks should be switched off or removed from their bedroom at night. They should be awakened softly.

Patients with LQT3 do not benefit from specific recommendations at the current time.

Molecular biology helps to better understand the relationship between ion channel gene mutation and patient symptoms.

Mutation with loss of function of potassium channel Kv7.1 coded by KCNQ1 and potassium channel Kv11.1 coded by KCNH2 lead to increased duration of the action potential of the myocardial cell and to prolonged QT interval duration. The activity of these channels that is physiologically increased by the sympathetic activity is central for the adaptation of the repolarization duration in case of tachycardia. Defect ion channels do not shorten repolarization and create a highly arrhythmogenic condition.

On the other hand, mutation with a gain of function of the sodium channel Nav1.5 coded by SCN5A prolongs the action potential of the myocardial cell membrane by increasing late sodium currents.

### 3.6. LQTS and Epilepsy

The clinical course of our patient and a previous case published [10] point out the causative relationship between LQTS with consecutive ventricular arrhythmias, decreased cerebral blood flow and cerebral seizures, misinterpreted as epilepsy.

However, it has been postulated that in the context of particular LQTS mutations, epilepsy might not be exclusively from cardiac origin but the manifestation of a pathogenic cardio-cerebral gene co-expression [20]. In such a case, however, the occurrence of cerebral seizures driven by the brain expression of the gene mutation might be independent of β-blocker prophylaxis.

Independently of their genotype, all patients with LQTS must avoid drugs prolonging the QT interval [2,4] www.qtdrugs.org (accessed on 26 February 2022), accessed on 26 February 2022 [11], hypokalemia, hypomagnesemia and some nutriments such as grapefruits [6].

### 3.7. Asymptomatic Patients

Asymptomatic patients with LQTS must be on β-blocker prophylactics. This is justified by the fact that about 10% of the patients will experience sudden cardiac death as the first clinical manifestation of LQTS. β-blocker prophylactics are also indicated in patients with normal QT interval (who represent 25% of all LQTS patients) in whom a pathogenic gene

mutation related to LQTS has been shown, regardless of their lower risk for sudden cardiac death [2,11].

No treatment is recommended in patients without LQTS-related mutation a first-degree relative of whom presents a gene mutation related to LQTS.

## 4. Conclusions

LQTS is a rare condition that may manifest clinically by malignant ventricular arrhythmias. Generalized cerebral seizures may be the unique mode of presentation consecutive to the decrease in the cerebral blood flow during ventricular arrhythmia. Physicians caring for children with refractory epilepsy should assure the exclusion of malignant arrhythmia secondary to LQTS.

Diagnosis of LQTS requires the selection of patients with a high probability to show the disease on the basis of their history, and clinical and electrophysiological findings in which genetic testing will be performed.

Genetic assessment is mandatory given the prognostic value of the gene mutation identified and the expected efficacy of the treatment proposed.

Non-cardio-selective β-blockers are the first choice of treatment for the prophylactics of adverse events that are always related to ventricular fibrillation and may end up in sudden cardiac death. If necessary, the treatment must be implemented by ICD implantation and/or LCSD.

**Author Contributions:** Conceptualization, J.L. and M.-C.S.; methodology, J.L., L.V.C. and S.B.; software, N.F. and J.L.; validation, M.-C.S., S.B. and L.V.C.; formal analysis, J.L.; investigation, J.L., N.F., S.B. and L.V.C.; resources, J.L.; data curation, J.L., N.F., S.B., L.V.C. and M.-C.S.; writing—original draft preparation, J.L.; writing—review and editing, L.V.C., S.B. and N.F.; visualization, N.F.; supervision, M.-C.S.; project administration, M.-C.S.; funding acquisition, None. All authors have read and agreed to the published version of the manuscript.

**Funding:** This research received no external funding.

**Institutional Review Board Statement:** Not applicable.

**Informed Consent Statement:** Not applicable.

**Data Availability Statement:** Not applicable.

**Conflicts of Interest:** The authors declare no conflict of interest.

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
