# Peer review of "Cerebral Seizures in an Adolescent with Jervell and Lange-Nielsen Syndrome: It May Not Be Epilepsy"

_clinpract, doi:10.3390/clinpract12050070_

Round 1
Reviewer 1 Report
Title: Cerebral seizures in an adolescent with Jervell and Lange-Nielsen syndrome: It may not be epilepsy
The paper is describing the case of cerebral seizures in a 13 years-old girl with Jervell and Lange-Nielsen syndrome (JLN). Most of the MS parts are well-balanced and well-written. However, some corrections and clarification are need prior to publishing.
Line 73: Please correct the spelling of the word “Jerwell” to “Jervell”.
Line 90: Please clarify the starting dose with propranolol.
Line 93: Please clarify the starting dose with phenobarbital therapy. Also, please clarify when the phenobarbital treatment has been stopped in the patient.
Line 110: Section 3 is missing or the numbering must be adjusted for the Discussion section.
Please cite the “Table 1” in the text as well.
Line 297 and 298: It is recommended to use “potassium channel” instead of “kalium channel”.
Also, I recommend that the authors cite and discuss this paper as well: " Goyal JP, Sethi A, Shah VB. Jervell and Lange-Nielson Syndrome masquerading as intractable epilepsy. Ann Indian Acad Neurol. 2012 Apr;15(2):145-7. doi: 10.4103/0972-2327.95003.”
Author Response
Dear Reviewer,
thank you very much for your important comments the answer to which is as follows:
- Line 73: Jervell is corrected
- Line 90: dose of propranolol is specified.
- Line 93: starting dose of phenobarbital is specified.
- Section 3 has been corrected.
- Table 1 is cited in the text.
- Potassium replaces kalium.
- The paper by Goyal et al. is now cited and discussed. We apologize for this omission and thank you very much for the advice.
We thank you again for your comments and hope to have answeerd all points adequately.
Sincerely yours
M-C Seghaye
Reviewer 2 Report
This case of epilepsy and syndrome with QT prolongation is very interesting. We would love to publish it in this journal, but more careful medical information needs to be added.
The abstract and key words are good.
I would like to see a little more detail in the introduction section, with additional background on how you came up with the idea of reporting this case.
The way the family tree is described is also correct.
In case presentation, I would like to request a chest radiograph. We would like to add that cardiac enlargement and cardiac shadowing were not a problem. Photos after pacemaker implantation are preferred.
Please provide additional information on whether there was or was not a need to limit daily exercise in this case, along with the reason.
How many EEGs were performed in total? How long was each recording?
(In the discussion section, please describe the criteria for the diagnosis of epilepsy when the EEG is negative, or the rate at which abnormalities are detected in the EEG test for the diagnosis of epilepsy.)
Has a brain CT, MRI, or MRA scan been performed because the EEG was not abnormal? Have you attempted to detect an epileptic focus by nuclear medicine or FDG-PET scan?
If this or a similar patient has a seizure, please describe in more detail how you would differentiate whether it was caused by a fatal arrhythmia or an epileptic seizure.
Also, while the cardiac medication management is very well described, please add a separate section on epilepsy control and any recommended antiepileptic medications.
Does the disease necessarily cause lethal arrhythmias and epileptic seizures? Do you know the penetration rate?
Overall, this paper contains important findings.
We hope that it will clear major revisions and be published in this journal.
Author Response
Dear Reviewer,
thank you very much for your useful comments the answer to which is as follows:
- Thank you for the positive comment on the abstract and key words.
- the introduction has been completed by the justification to share this kind of case with our community.
- Thank you for the positive comment on the family tree.
- We now add a chest X-Ray. Please consider that the child had several cardiac examination and that a structural cardiac defect was excluded from the very begining on. The child received an event recorder implanted sub-cutaneously and not a pace-maker or defibrilator.
It is not our policy to make a chest X-ray after this kind of intervention.
However, the chest X-ray shown here is normal and is useful to inform the reader about the absence of a significant structural heart defect. - The advice about the necessity to limit sports activity has now been added. Usually, given that the patients are on b-blockers, such a high level of exercice tolerance cannot be achieved in children. Nevertheless, our policy is to clearly inform parents about the interdiction/impossibility for their child to participate to high level sports activities. This is now mentionned in the text and supports the later discussion on that important point. Thank you very much for this advice.
- The number of EEGs performed is now mentioned. It is also specified that at least a prolonged night EEG was recorded in adition to the standard ones.
It is now specified in the discussion why the diagnosis of epilepsy was challenged. - A cerebral CT-scan was performed that was normal.
- Today, given the possibility to place sub-cutaneous event recorder even in small children, we would always propose to the parents this technology that gives clear answer about the presence of arrhythmias in children who present unexplained seizures. This is mentioned in the discussion (sentences 216-230).
- We have added a paragraph on LQTS and epilepsy, the treatment of which is well established by the neuropediatric community.
- There are no specific recommandation on the treatment of seizures secondary to cardiac arrhythmias but the anti-arrhythmic prophylaxis.
Your comment stimulated us however to redo a literature search on the link between epilepsy and arrhythmias and we found an interesting paper that suggests the presence of a co-expression in the heart and in the brain in patients with specific LQTS mutations. This remains speculative but important enough, as we feel, to be commented.
A new paragraph LQTS and epilepsy has been added in the discussion section.
The fatal issue of tosades de pointes and ventricular fibrillation in LQTS is pointed out in the text because sudden cardiac death may be the only manifestation of LQTS.
We tahnk you for your comments and hope having answered all points adequately.
Sincerely yours
M-C Seghaye
Round 2
Reviewer 2 Report
Thank you for your re-submission to this journal.
I believe that you have revised the content to adequately address my difficult question and comment with the addition of two references. Thank you also for the minor corrections that I did not notice. This paper is now a high quality case report that includes a comprehensive description of LQT.
I would rate the content as sufficient to warrant publication.
I look forward to your further submissions to our journal.
Best regards, Reviewer